# TEACHER-STUDENT COMPRESSION WITH GENERATIVE ADVERSARIAL NETWORKS

## ABSTRACT

More accurate machine learning models often demand more computation and memory at test time, making them difficult to deploy on CPU- or memory-constrained devices. *Teacher-student compression (TSC)*, also known as *distillation*, alleviates this burden by training a less expensive student model to mimic the expensive teacher model while maintaining most of the original accuracy. However, when fresh data is unavailable for the compression task, the teacher's training data is typically reused, leading to suboptimal compression. In this work, we propose to augment the compression dataset with synthetic data from a generative adversarial network (GAN) designed to approximate the training data distribution. Our *GAN-assisted TSC* (GAN-TSC) significantly improves student accuracy for expensive models such as large random forests and deep neural networks on both tabular and image datasets. Building on these results, we propose a comprehensive metric—the *TSC Score*—to evaluate the quality of synthetic datasets based on their induced TSC performance. The TSC Score captures both data diversity and class affinity, and we illustrate its benefits over the popular Inception Score in the context of image classification.

## 1 INTRODUCTION

Modern machine learning models have achieved remarkable levels of accuracy, but their complexity can make them slow to query, expensive to store, and difficult to deploy for real-world use. Ideally, we would like to replace such cumbersome models with simpler models that perform equally well. One way to address this problem is to perform *teacher-student compression* (*TSC*, also known as *distillation*), which consists of training a student model to mimic the outputs of a teacher model (Bucila et al., 2006; Li et al., 2014; Hinton et al., 2015). For example, expensive ensemble and deep neural network (DNN) teachers have been used to train inexpensive decision tree (Craven & Shavlik, 1996; Frosst & Hinton, 2017) and shallow neural network (Bucila et al., 2006; Li et al., 2014; Ba & Caruana, 2014; Hinton et al., 2015; Urban et al., 2017) students. While alternative model-specific compression strategies abound (see Section 6), TSC is distinguished by its broad applicability: the same framework can be used to compress any classifier, be it a random forest or a deep neural network.

An important degree of freedom in the TSC problem is the *compression set* used to train the student. Ideally, fresh (unlabeled) data from the training distribution would fuel this task, but often no fresh data remains after the teacher is trained (Bucila et al., 2006; Ba & Caruana, 2014). In this case, one branch of the literature, dating back to the pioneering work of Bucila et al. (2006), recommends generating synthetic data for compression and proposes tailored generation schemes for tabular (Bucila et al., 2006) and image (Urban et al., 2017) data. A second branch, rooted in the distillation community (Hinton et al., 2015; Frosst & Hinton, 2017), simply uses the same data to train teacher and student (see also Ba & Caruana, 2014). Here, we show that the latter convention leads to suboptimal compression performance and propose a synthetic data generation strategy for both tabular and image data that improves upon standard augmentation schemes. Specifically, when fresh data is unavailable for TSC, we propose to augment the compression set with synthetic data produced by generative adversarial networks (GANs) (Goodfellow et al., 2014). GANs attempt to generate new datapoints from the distribution underlying a given dataset and have achieved impressive fidelity for a variety of data types including images (Goodfellow et al., 2014), text (Yu et al., 2017), and electronic health records (Choi et al., 2017). Here, we identify TSC as a practical downstream task

for which GAN generation is consistently useful across data types and classification tasks and develop *GAN-assisted TSC* (GAN-TSC) to improve the TSC of an arbitrary classifier. Our extensive empirical evaluation demonstrates the effectiveness of GAN-TSC for tabular data (for which GANs are seldom used), image data, random forest classifiers, and DNN classifiers.

**Why should synthetic data improve TSC?** Note that there is an important distinction between training a student to mimic a teacher with synthetic data and training a student to solve the original supervised learning problem with synthetic data. The goal of the original supervised learning task is to approximate the ideal mapping $f^*$ between inputs $x$ and outputs $y$. This ideal $f^*$ is a functional of the true but unknown distribution underlying our data, and our information concerning $f^*$ is limited by the real data we have collected. The goal in TSC is to approximate the teacher prediction function $g$ which maps from inputs to predictions $z$. Because the teacher is a function of the training data alone, $g$ itself is a functional of the training data alone and is otherwise independent of the unknown distribution that generated that data. In addition, because we have access to the teacher, we have the freedom to query the function $g$ at any point, and hence our information concerning $g$ is limited only by the number of queries we can afford. In particular, when we generate a new query point $x$, we can observe the actual target value of interest, the teacher's prediction $g(x)$; this is not true for the supervised learning task, where no new labels can be observed. The insensitivity to errors in synthetic labels and access to fine-grained teacher predictions make TSC more ideally suited to synthetic data augmentation. Indeed, we will see in Sections 3 and 4 that the same GAN data that leads to improved TSC leads to degraded accuracy when used to augment the original supervised learning training set. This is consistent with past work that demonstrates gains from GAN-augmented supervised learning in specific data-starved situations but reports degraded accuracy when all training data is used (Bowles et al., 2018, Tab. 4). See (Ba & Caruana, 2014) for further discussion on the distinctions between TSC and the original supervised learning task.

Since the improvement realized by GAN-TSC depends on the synthetic data quality, we further propose to use GAN-TSC to evaluate the quality of synthetic datasets and their generators. In essence, we declare a synthetic dataset to be of higher quality if a compressed model trained on that data achieves higher test accuracy. Synthetic data evaluation is a notoriously difficult problem marked by the lack of universally agreed-upon quality measures (Theis et al., 2015). Some standard quality measures, like *multiscale structural similarity* (Wang et al., 2003), quantify the diversity of a synthetic dataset but do not capture *class affinity*, the ability of datapoints to be correctly associated with their labels with high confidence. Others, like the popular *Inception Score* (Salimans et al., 2016), quantify class affinity based on the predicted label distribution of a trained neural network. However, these scores do not account for within-class diversity and are easily misled by adversarial datapoints that elicit high confidence predictions but do not resemble real data. To address these shortcomings, we develop a *TSC Score* that quantifies the true test accuracy of compressed models trained using synthetic data; this offers a robust, goal-driven metric for synthetic data quality that accounts for both diversity and class affinity. In summary, we make the following principal contributions:

1. We identify TSC as a practical downstream task for which GAN data augmentation is consistently useful across data types and classification tasks and develop GAN-TSC as a drop-in replacement for standard TSC.

2. For random forest teachers, we demonstrate 25 to 336-fold reductions in execution and storage costs with less than $1.2\%$ loss in test performance across a suite of real-world tabular datasets. In each case, GAN-TSC improves over student training without TSC, over TSC without synthetic data augmentation, and over the tabular data augmentation strategy of Bucila et al. (2006).

3. For image classification, we show GAN-TSC consistently improves student test accuracy for a variety of deep neural network teacher-student pairings and two popular compression objectives.

4. We introduce a new TSC Score for evaluating the quality of GAN-generated datasets and, on Caltech-256 and CIFAR-10, illustrate its value and advantages over the popular Inception Score.

## 2 TEACHER-STUDENT COMPRESSION WITH GANS

We begin by reviewing standard approaches to DNN TSC and describing our proposals for random forest TSC and improving TSC with GAN data.

**Deep Neural Network TSC**   In the standard teacher-student approach to compressing a neural network classifier, a relatively inexpensive prediction rule, like a shallow neural network, is trained to predict the unnormalized log probability values—the *logits z*—assigned to each class by a previously trained deep network classifier. The inexpensive model is termed the *student*, and the expensive deep network is termed the *teacher*. Given a compression set of $n$ feature vectors paired with teacher logit vectors, $\{(x^{(1)}, z^{(1)}), ..., (x^{(n)}, z^{(n)})\}$, Ba & Caruana (2014) proposed framing the TSC task as a multitask regression problem with $L^2$ loss, $L(\theta) = ||g(x; \theta) - z||_2^2$. Here, $\theta$ represents any student model parameters to be learned (e.g., the student network weights), and $g(x; \theta)$ is the vector of logits predicted by the student model for the input feature vector $x$.

Li et al. (2014) introduced an alternative TSC objective function, and Hinton et al. (2015) parameterized this objective by a temperature parameter $T > 0$. Specifically, the student is trained to mimic the annealed teacher class probabilities, $q_j(z/T) = \exp(z_j/T)/\sum_k \exp(z_k/T)$, for each class $j$ by solving a multitask regression problem with cross-entropy loss, $L_T(\theta) = -\sum_j q_j(z/T) \log(q_j(g(x; \theta)/T))$. Hinton et al. (2015) showed that, under a zero-mean logit assumption, cross-entropy regression recovers $L^2$ logit matching as $T \to \infty$; however, the two approaches can differ for small $T$. In Sec. 4, we will experiment with both of these popular TSC approaches.

**Random Forest TSC**   Random forests (Breiman, 2001) construct highly accurate prediction rules by averaging the predictions of a diverse and often large collection of learned decision trees. Effectively mimicking a large random forest with a single decision tree or a small forest has the potential to reduce prediction computation and storage costs by multiple orders of magnitude (Bucila et al., 2006; Joly et al., 2012; Begon et al., 2017; Painsky & Rosset, 2016; 2018). Focusing on the common setting of binary classification, we propose to train a student regression random forest to predict a teacher forest's outputted probability $p$ of a datapoint $x$ having the label 1.

**Reducing overfitting with GAN-assisted TSC (GAN-TSC)**   In a typical TSC setting, as much data as possible has been dedicated to training the highly accurate teacher model, leaving little fresh data for training the student model. While one branch of the TSC literature recommends generating synthetic data with customized augmentation algorithms for tabular (Bucila et al., 2006) and image (Urban et al., 2017) data, the more common solution in the distillation literature is to simply reuse the teacher training set as the compression set (Hinton et al., 2015; Frosst & Hinton, 2017). However, we will see in Secs. 3 and 4 that compressing with training data alone leads to suboptimal student performance. This suboptimality occurs both due to teacher overfitting (many teachers perform very well on test data but are still overfit in the sense of having unrealistically small training error or overconfident training logits not representative of its test logits; these are the teachers that benefit most from GAN-TSC) and student overfitting (the student can benefit from observing the teacher's outputs at points other than the original training points). To boost student performance and compression efficiency, we propose a simple solution applicable to tabular and image data alike: augment the compression set with synthetic feature vectors generated by a high-quality GAN. These synthetic feature vectors are then labeled with the teacher's outputted class probabilities or logits. We call this approach *GAN-assisted TSC* and release our Python implementation at `REDACTED`.

**AC-GAN**   To generate high-quality GAN feature vectors which capture the salient features of each class, we use the auxiliary classifier GAN (AC-GAN) of Odena et al. (2017). The AC-GAN generator $G$ produces a synthetic feature vector $X_{fake} = G(W, C)$ from a random noise vector $W$ and an independent target class label $C$ drawn from the real data class distribution. For any given feature vector $x$, the AC-GAN discriminator $D$ predicts both the probability of each class label $P(C \mid x)$ and the probability of the data source being real or fake, $P(S \mid x)$ for $S \in \{real, fake\}$. For a given training set $\mathcal{D}_{real}$ of labeled feature vectors, two components contribute to the AC-GAN training objective,

$$L_{source} = \frac{1}{|\mathcal{D}_{real}|} \sum_{(x,c) \in \mathcal{D}_{real}} \log P(S = real \mid x) + \mathbb{E}_{W,C \sim p_c}[\log P(S = fake \mid G(W, C))] \text{ and}$$
$$L_{class} = \frac{1}{|\mathcal{D}_{real}|} \sum_{(x,c) \in \mathcal{D}_{real}} \log P(C = c \mid x) + \mathbb{E}_{W,C \sim p_c}[\log P(C \mid G(W, C))], \tag{1}$$

representing the expected conditional log-likelihood of the correct source and the correct class of a feature vector, respectively. Training proceeds as an adversarial game with the generator $G$ trained to maximize $L_{class} - L_{source}$ and the discriminator $D$ trained to maximize $L_{class} + L_{source}$.

## 3 RANDOM FOREST GAN-TSC

We now explore how GAN-TSC performs when used to compress large random forests for binary classification. We employ three real-world tabular datasets. The MAGIC Gamma Telescope dataset (Dheeru & Karra Taniskidou, 2017) task is to distinguish hadronic showers from primary gamma signals recorded by a gamma telescope; $64.8\%$ of datapoints have the label 0 (signal). We select a uniformly random subset of 200,000 class-balanced datapoints from the Higgs dataset (Dheeru & Karra Taniskidou, 2017) to predict whether a given observation was produced by a Higgs boson. Following the feature extraction protocol of (Liu et al., 2017), we extract 29 continuous features from the StumbleUpon Evergreen dataset Eve to predict whether a given web page is evergreen; $48.7\%$ of datapoints have the label 1. We split each dataset into training and test sets uniformly at random, with training split sizes given in Figs. 1a-1d.

In our experiments, the teacher is a random forest classifier with 500 trees, and the student is a regression random forest with one to 20 trees; both are trained using scikit-learn (Pedregosa et al., 2011) with default values for all the hyperparameters. For the AC-GAN implementation in Keras, both the generator and the discriminator are one layer fully-connected neural networks with 50 neurons and ReLU activation. We employed noise vectors $w \in \mathbb{R}^{100}$ and an Adam optimizer with learning rate 0.0002 and momentum term $\beta_1 = 0.5$.

We study three scenarios: TSC using training data only, GAN data only or a mixture of training and GAN data. We generate $n_{fake} = 9\,n_{real}$ GAN datapoints for the compression set, where $n_{real}$ is the number of real training datapoints. The mixture compression set is generated by pooling the $n_{real}$ training datapoints and the $n_{fake}$ GAN datapoints together. We also report the performance of a student trained directly on the original training set without TSC ('Student Only'); since the student is a regression forest, the class labels (0 and 1) are treated as real value targets.

The results of compressing a random forest with 500 trees into one or more decisions trees are given in Figs. 1a-1d. We experiment with a variety of training dataset sizes, ranging from $n = 1\text{k}$ to $n = 100\text{k}$ to demonstrate the versatility of GAN-MC. In each case, the trees trained by the teacher and students have similar depth after training. We use test accuracy as our performance metric for the balanced Higgs dataset and test AUC for the unbalanced MAGIC and Evergreen datasets. For all datasets, TSC into a single tree with GAN data outperforms TSC with training data and substantially outperforms the student model trained without TSC. Moreover, for the Higgs dataset, the accuracy boost from GAN-TSC ($62.1\%$ to $69.6\%$ on Higgs 100k) is 10 times the accuracy boost achieved using training data TSC ($62.1\%$ to $62.7\%$).

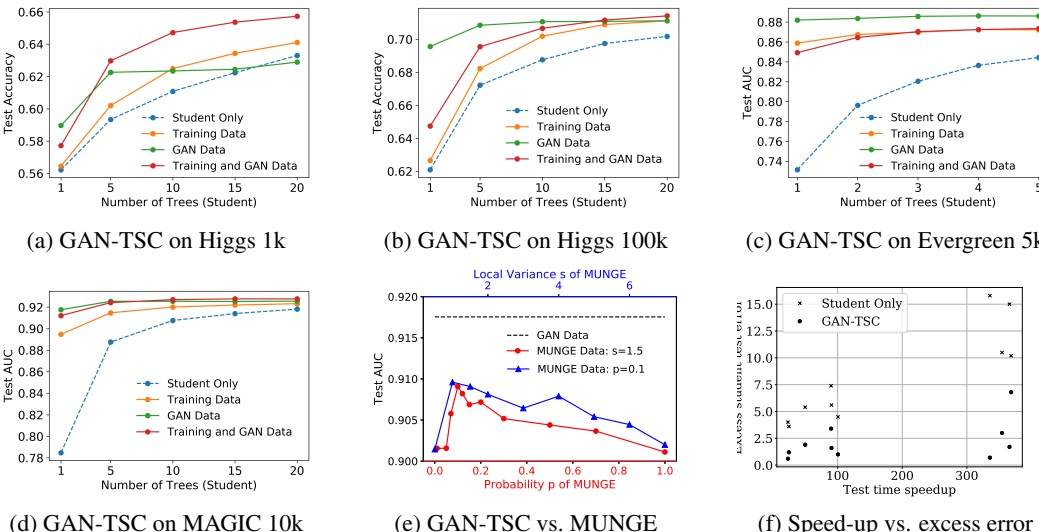

Figure 1: (a-e) Student test accuracy (Higgs) and test AUC (Evergreen and MAGIC) when compressing a 500 tree random forest into a compact forest. (f) For all datasets, GAN-TSC students increase test time throughput 25 to 336-fold over teacher with less than $1.2\%$ loss of accuracy.

The example of the Evergreen dataset is also enlightening. Compression into a single tree with training data increases student test AUC from 0.731 to 0.856, and compression with only GAN data yields a further improvement to 0.882, nearly matching the 0.889 test AUC of the teacher. Remarkably, this is achieved with a single decision tree which demands 336 times less computation and storage space than the teacher at prediction time. The figure 336 comes from an assessment of student test-time speed-ups summarized in Fig. 1f. For each dataset, we identified the highest accuracy and most compressed students trained with and without GAN-TSC and measured throughput as the time needed to compute predictions for $30,000$ test examples using one core of an Intel Xeon 6152 processor. At a cutoff of $1.2\%$ excess test error, we observe speed-ups ranging from 25 to 336-fold.

For each dataset save Higgs 1k, TSC with GAN data offers the best (or nearly the best) performance for all forest sizes. For the Evergreen and MAGIC datasets, near-maximal performance is achieved by a single GAN-TSC decision tree, with additional trees yielding relatively minor AUC gains. For Higgs 1k, the combination of training and GAN data offers the best performance for all multi-tree forests, with an accuracy boost consistently 2-4 times that of TSC with training data alone.

**GAN-TSC vs. MUNGE.** For tabular data, an alternative to GAN-TSC is to augment the compression set in precisely the same way with data generated from the state-of-the-art tabular augmentation strategy, MUNGE, of (Bucila et al., 2006). We find that, for all datasets (Fig. 1e displays a representative comparison for a single decision-tree student on MAGIC; additional comparisons are made in Appendix A.2) and all settings of the MUNGE hyperparameters (the local variance $s$ and the probability $p$ optimally tuned on the test set in Fig. 1e), MUNGE underperforms GAN-TSC.

## 4 DEEP NEURAL NETWORK GAN-TSC

We now investigate how GAN-TSC performs when used to compress convolutional DNN (CNN) classifiers trained on the CIFAR-10 dataset of (Krizhevsky & Hinton, 2009). CIFAR-10 consists of $32 \times 32$ RGB images from 10 classes, divided into 50,000 training and 10,000 test images. The test images are randomly divided into a validation set with size 5000 and a test set with size 5000. The AC-GAN is implemented in Keras (Chollet et al., 2015) and trained for 1000 epochs (Tuya, 2017). The discriminator $D$ is a CNN with 6 convolution layers and Leaky ReLU nonlinearity. The generator $G$ consists of 3 'deconvolution' layers which transform the class $c$ and noise vector $w \in \mathbb{R}^{110}$ into a $32 \times 32$ image with 3 color channels. We use the Adam optimizer with learning rate 0.0002 and momentum term $\beta_1 = 0.5$, as suggested by (Radford et al., 2015).

We employ both of the TSC objectives introduced in Sec. 2 using 200 TSC training epochs. For $L^2$ logit matching, the teacher and the student are NIN (Lin et al., 2014) and LeNet (LeCun et al., 1998) models. The uncompressed networks are pre-trained by Caffe (Chan, 2016; Jia et al., 2014). Similar to (Chan, 2016), for TSC training, we use the Adam optimizer in Tensorflow (Abadi et al., 2015) with $L^2$ loss and learning rate $10^{-4}$.

For cross-entropy regression, we examine three additional networks: WideResNet-28-10 (Zagoruyko & Komodakis, 2016), ResNet-18 (He et al., 2016), and a 5-layer CNN with 3 convolution layers. Network training both with and without compression is carried out in Pytorch (Li, 2018; Paszke et al., 2017). For TSC, we use the student objective $L(\theta) = \alpha L_T(\theta) + (1-\alpha)L_0(\theta)$, where $\alpha \in (0,1]$ and $L_0(\theta) = -\sum_j \mathbf{1}\{j = c\}\log(q_j(g(x;\theta)))$ is the cross-entropy classification loss for a datapoint $x$ with class label $c$. For each teacher-student pair, we set $T$, $\alpha$, and all optimizer hyperparameters to the default values recommended in (Li, 2018). For the teacher-student pairs 1, 2, and 3 in Table 1, this yields the respective $T$ values 5, 20, and 6 and $\alpha$ values 0.9, 0.9, and 0.95. The Adam optimizer with learning rate $10^{-3}$ is used for teacher-student pairs 1 and 2, and stochastic gradient descent with learning rate decayed from 0.1 is used for pair 3.

We compare the standard approach of TSC using only the teacher's training dataset to two versions of GAN-TSC: compression using only GAN data and compression using a mixture of training and GAN data. The GAN data is produced in real time during the stochastic optimization training. The mixture of training and GAN data is realized by generating GAN data with probability $p_{fake}$ and by sampling from the training set with probability $1 - p_{fake}$. For each teacher-student pair, we select the value of $p_{fake}$ in $\{0.0, 0.1, 0.2, \ldots, 1.0\}$ that yields the highest validation set accuracy and report performance on the held-out test set. This results in the choice $p_{fake} = 0.8$ for the NIN-LeNet teacher-student pair and 0.2 for the other pairings.

Fig. 2a displays student test accuracy following each epoch of TSC training with the $L^2$ logit-matching objective. In the end, both versions of GAN-TSC significantly outperform TSC on training data alone and training without TSC ('Student Only'). The results are particularly striking for the mixture of GAN and training data which doubles the impact of training data TSC. In this case, student accuracy increases by 10.5 percentage points (from 66.2% to 76.7%) with GAN-TSC as opposed to 5.3 percentage points (from 66.2% to 71.5%) with training data alone. Table 1 reports comparable improvements for the NIN-LeNet teacher-student pairing when the cross-entropy TSC objective is used. Indeed, the mixture of GAN and training data improves upon training data TSC for all teacher-student pairings investigated.

At the start of the TSC training in Fig. 2, TSC with training data is most effective, presumably because the real training data provide a more faithful reflection of the test data distribution, and the overfitting effect is not yet severe. Correspondingly, a quicker increase in test accuracy is observed at the start in Fig. 2a. After approximately 10 epochs, the influence of overfitting gradually increases and becomes dominant over the advantage of fidelity to the test data distribution. The compression set loss for real training data becomes significantly smaller than the loss with either version of GAN-TSC in Fig. 2b, and the test accuracy stops increasing in Fig. 2a. Moreover, the teachers in our experiments yield 100% accuracy on the training set but significantly lower accuracy on test datapoints, indicating a significant difference between the distributions of training and test set logit values and a disadvantage to relying wholly on training points. This dynamic illustrates the trade-off between GAN faithfulness to the real data distribution and the influence of overfitting and suggests that GAN-TSC improves accuracy by mitigating overfitting to the compression set using a plentiful source of fresh and realistic (albeit imperfect) data.

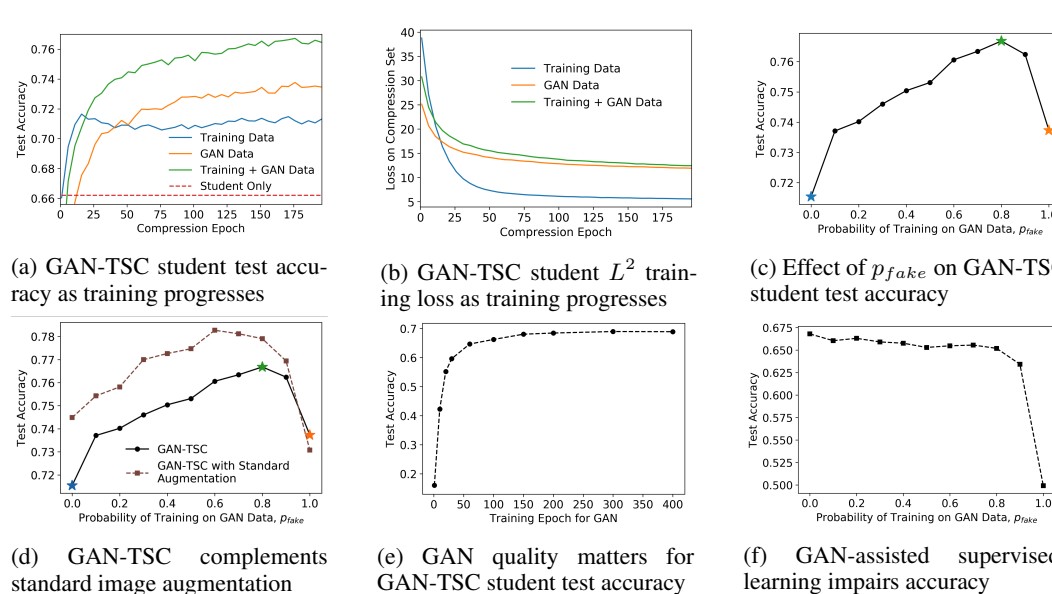

(a) GAN-TSC student test accuracy as training progresses

(b) GAN-TSC student $L^2$ training loss as training progresses

(c) Effect of $p_{fake}$ on GAN-TSC student test accuracy

(d) GAN-TSC complements standard image augmentation

(e) GAN quality matters for GAN-TSC student test accuracy

(f) GAN-assisted supervised learning impairs accuracy

Figure 2: Student performance for $L^2$ logit-matching compression on CIFAR-10 (see Sec. 4), averaged over 3 independent student training runs. The teacher and student are NIN and LeNet with test accuracies of 78.1% and 66.2% (red dashed curve in (a)) when trained without compression.

Table 1: CIFAR-10 image classification test accuracies for cross-entropy compression with various teacher and student neural net architectures. GAN-TSC outperforms standard TSC on training data alone and training without compression ('Student Only') in all cases. See Sec. 4 for more details.

| | Teacher | Student | Teacher Only | Student Only | Student after Compression with | |
| --- | --- | --- | --- | --- | --- | --- |
| | | | | | Training Data | Training & GAN |
| 1 | NIN | LeNet | 78.1% | 66.2% | 71.0% | **75.3%** |
| 2 | ResNet-18 | 5-layer CNN | 94.2% | 78.8% | 84.4% | **86.6%** |
| 3 | WideResNet-28-10 | ResNet-18 | 95.8% | 94.2% | 94.3% | **95.0%** |

**Effect of the GAN training proportion parameter** $p_{fake}$**.** Adopting the experimental setup of Fig. 2, we next examine how $p_{fake}$, the probability of selecting a GAN datapoint over a real datapoint when training the student, affects compression performance. We plot the dependence of trained student test accuracy on $p_{fake}$ in Fig. 2c. When $p_{fake} = 0$, only training data is used for compression; when $p_{fake} = 1$, only GAN data is used. Notably, every non-zero setting of $p_{fake}$ leads to improved accuracy over compression with the real training data alone, underscoring the value of GAN augmentation. Beyond this, we observe a non-monotonic but unimodal dependence on $p_{fake}$ with a combination of GAN and real datapoints providing significantly higher accuracy than GAN or real datapoints alone. This is consistent with a trade-off between the overfitting caused by training data reuse and the inability of a GAN to perfectly approximate the true data distribution.

**GAN-TSC complements standard augmentation.** Our next experiment explores the impact of standard image augmentation on compression with and without GAN-TSC. We adopt the experimental setup of Fig. 2 but, during teacher and student training, we introduce the random image augmentations, in the form of left-right image flips and hue-saturation-value (HSV) shifts and scalings, responsible for the state-of-the-art TSC performance in (Urban et al., 2017). In Fig. 2d, we see that a student compressed with standard augmentation alone has 74.5% test accuracy (versus 76.7% for GAN-TSC without standard augmentation); however, the greatest gain is realized when GAN and standard augmentation are combined, yielding a maximum accuracy of 78.3%.

**GAN quality matters.** To investigate the degree to which synthetic data quality affects TSC improvement, we repeat the experiment of Fig. 2 using GAN data of varying quality and $p_{fake} = 1$. We use the number of GAN training epochs as a proxy for GAN quality. In Fig. 2e, we see student test accuracy is greatly impaired by using a low-quality GAN trained for too few epochs. Fortunately, student accuracy monotonically improves as the number of epochs and GAN fidelity increase.

**GAN-TSC vs. GAN-assisted supervised learning.** In Sec. 1, we discussed the significant differences between GAN-TSC and using GAN data to augment the training set for the original supervised learning problem. Fig. 2f shows that the same mixtures of GAN and training data that improve student compression performance in Fig. 2c actually impair accuracy when the student is trained without compression for the original supervised learning task. We observe the same phenomenon in random forest compression (see Fig. 4 in the supplement).

## 5    A TEACHER-STUDENT COMPRESSION SCORE FOR EVALUATING GANS

The evaluation of synthetic datasets is an important but challenging task. Two criteria commonly considered essential for a high-quality synthetic dataset are datapoint diversity and class affinity. The most widely used GAN quality measure, the Inception Score (IS) of Salimans et al. (2016), measures across-class diversity but does not account for within class diversity. In addition, the IS measures a form of class affinity based on the predictions of a pre-trained neural network but is easily misled by datapoints that elicit high confidence predictions without resembling real data. For example, if the classification loss $L_{class}$ is heavily upweighted relative to the source loss $L_{source}$ while training an AC-GAN, the generator will be more likely to produce feature vectors classified with high confidence by neural networks. As we will see in Sec. 5, such feature vectors need not resemble real data but will nevertheless receive high ISs (which should be reserved for high-quality datasets). To account for both class affinity and diversity in a more robust and holistic manner, we propose to use the performance of a student trained on GAN data as a measure of GAN dataset quality. For reproducibility, Python code to compute the TSC Score is available at   REDACTED.

**The TSC Score**   To evaluate the quality of a generated dataset $\mathcal{D}$ relative to a real dataset $\mathcal{D}_{real}$, we define a *Teacher-Student Compression Score (TSCS)* based on the test accuracy $\mathrm{acc}(\mathcal{D})$ of a student trained with compression set $\mathcal{D}$ to mimic a pre-trained teacher:

$$\mathbf{TSCScore}(\mathcal{D}; \mathcal{D}_{real}) = \frac{\mathrm{acc}(\mathcal{D}) - \mathrm{acc}_{\mathrm{mode}}}{\mathrm{acc}(\mathcal{D}_{real}) - \mathrm{acc}_{\mathrm{mode}}},$$

where $\mathrm{acc}_{\mathrm{mode}}$ is the accuracy obtained by always predicting the most common class in the test set. In our experiments, we choose $\mathcal{D}_{real}$ to be the teacher's training data, but any choice is equally valid, as the ranking induced by the TSCS is not affected by the choice of $\mathcal{D}_{real}$.

The TSCS declares a synthetic dataset to be of higher quality if a compressed model trained only on that data achieves higher accuracy on real test data. The score takes values in $[0, \infty)$ and tends

to 0 as the synthetic data distribution diverges from the real data distribution. Appealingly, the TSCS tends to increase in response to increases in within-class diversity, across-class diversity, and class affinity, as each can enable the student to more accurately mimic the teacher's output across all classes. This makes the TSCS a more holistic measure of synthetic data quality than the IS or multiscale structural similarity. However, crucially, the TSCS is only impacted by aspects of class affinity and diversity that matter for performance on real test data. Hence, unlike the IS which is completely determined by the idiosyncratic output of an imperfect network, the TSCS is robust to the idiosyncratic preferences of an imperfect teacher or student. In particular, we would not expect a student trained on unrealistic or adversarial synthetic data to perform well on real test data even if it very accurately mimics the teacher's predictions on such data. A potential inconvenience of the TSCS is the need to train an inexpensive student model. To ensure that the TSCS can be computed efficiently, we train each student for only one epoch; our experiments suggest that this is sufficient to effectively capture GAN data quality and can be less expensive then evaluating the IS.

**Scoring GANs: An Illustration with Caltech-256** As a first simple illustration of TSCS behavior on real and synthetic data, we use the Caltech-256 dataset (Griffin et al., 2007) with 256 object catogories and 30,607 total images. Because of the few samples per class and high within-class variability, Caltech-256 is a challenging dataset for GANs, and we would expect an AC-GAN trained to be of relatively low quality. To train our AC-GAN and perform evaluations, we randomly split the data into 100 training and 20 test images per category; notably, the fake images produced by the AC-GAN (see Fig. 3) are visually quite distinct from real Caltech-256 photos (Fig. 5 in the supplement). To compute the TSCS, we choose Xception (Chollet, 2017) as the teacher and SqueezeNet (Iandola et al., 2016) as the student. In line with our expectations,

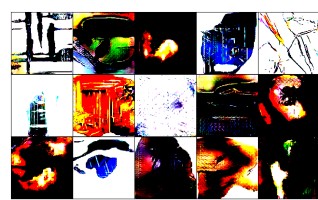

Figure 3: GAN data for Caltech-256.

the TSCS drops from $0.96 \pm 0.05$ (real images) to $0.01 \pm 0.01$ (fake images), indicating that little useful information could be learned from the low-quality GAN data. Comfortingly, in this example, the standard IS also decreases (from $21 \pm 1$ for real images to $8.0 \pm 0.1$ for fake images), but we will see next that the IS does not always behave as expected.

**TSC vs. Inception: An Illustration with CIFAR-10** To illustrate the benefit of the TSCS over the commonly-used IS, we reinstate the CIFAR-10 experimental setup of Fig. 2. We evaluate the TSCS on 50K CIFAR-10 images (the teacher's training data), 50K well-trained GAN images (i.e., data from the AC-GAN described in Sec. 4), and 50K inferior images which have high confidence classifications under the teacher network but do not resemble real data. The inferior data is generated by training the well-trained AC-GAN for 10 additional epochs using only the classification objective $L_{class}$ (1). That is, both the generator $G$ and discriminator $D$ are trained to maximize $L_{class}$, while ignoring the traditional GAN objective component $L_{source}$. We report means and standard errors across 3 independent runs. In Table 2, the GAN data quality degrades noticeably after the additional training with only $L_{class}$, and the TSCS decreases in accordance with our expectations. However, the IS increases for the inferior GAN images despite the evident unrealistic artifacts.

Table 2: Inception and TSC Scores for CIFAR-10 images; larger scores should signify higher quality. TSCS decreases for inferior images, but IS increases despite evident unrealistic artifacts.

| Real Data | Well-trained GAN | Inferior GAN |
|---|---|---|
| Inception: $11.2 \pm 0.1$ | Inception: $5.80 \pm 0.06$ | Inception: $5.93 \pm 0.06$ |
| Compression: $0.994 \pm 0.003$ | Compression: $0.778 \pm 0.002$ | Compression: $0.702 \pm 0.002$ |

To highlight the practicality of the TSCS, we also report a timing comparison of the IS and TSCS evaluations. To compute the IS, we perform one Inception network forward pass on 50K GAN images. To compute the TSCS, we first perform one forward pass on the same 50K images to get the NIN teacher's logits. We then train the LeNet student for one epoch with one forward and one backward pass. We finally perform one forward pass on 10K real test images to compute student test accuracy. Using the IS code of (Salimans et al., 2016) and an NVIDIA Tesla V100 GPU, the IS required 1436.6s and the TSCS 350.1s.

## 6 RELATED AND FUTURE WORK

To reduce the deployment costs of expensive machine learning classifiers, we introduced GAN-assisted TSC as a straightforward way to improve teacher-student compression. We demonstrated the benefits of GAN-TSC for both tabular and image data classifiers and developed a new TSC Score for evaluating the quality of synthetic datasets. While we have focused on improving the popular teacher-student paradigm of compression, we would be remiss to not mention alternative, model-specific approaches to reducing deployment costs, including parameter sharing (Chen et al., 2015), network pruning (Han et al., 2015), and network parameter prediction (Denil et al., 2013) for DNNs and indicator function selection (Joly et al., 2012), pre-pruning (Begon et al., 2017), and probabilistic modeling and clustering (Painsky & Rosset, 2016; 2018) for random forests.

A number of exciting opportunities for future work remain. For example, GAN-TSC is readily integrated into more complex TSC approaches that currently reuse the teacher's training data for compression. Prime examples are the recent approaches of (Wang et al., 2018b; Xu et al., 2018; Wang et al., 2018a). These differ from standard TSC by employing non-standard GAN-type compression losses, in which the student acts as the discriminator (Wang et al., 2018b) or generator (Xu et al., 2018; Wang et al., 2018a); Wang et al. (2018a) also train the teacher and student together. In addition, GAN development for tabular data has received much less attention than GAN development for image data, and we anticipate that significant improvements over the AC-GANs used in our experiments will result in significant performance benefits for GAN-TSC.

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

# A SUPPLEMENT

## A.1 RANDOM FOREST GAN-TSC VS. GAN-ASSISTED SUPERVISED LEARNING

Consistent with our discussion in Sec. 1 and our findings in Fig. 2f, Fig. 4 shows that the same GAN data that substantially improves student compression performance in Fig. 1d harms or scarcely improves test AUC when the random forest student is trained without compression for the original MAGIC supervised learning task.

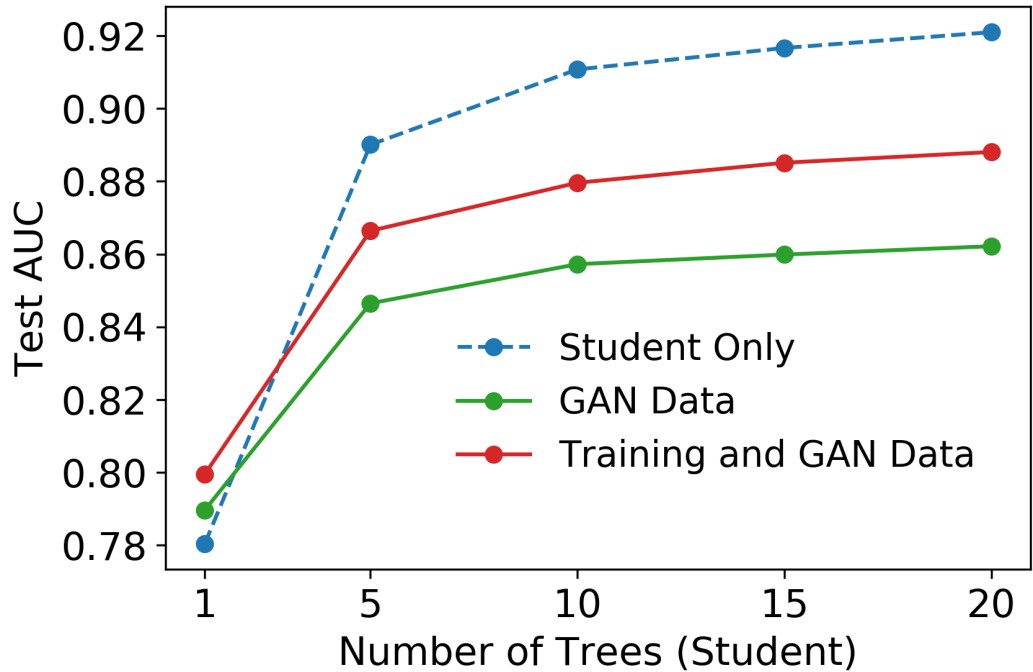

Figure 4: Without compression, a random forest student is trained for the original supervised learning task on MAGIC using only training data ('Student Only'), only GAN data, or a mixture of training and GAN data.

## A.2 RANDOM FOREST GAN-TSC VS. MUNGE

On the task of compressing a 500-tree random forest into a single tree (see Sec. 3), GAN-TSC outperforms TSC with MUNGE data on every dataset ((Higgs 100k: MUNGE 64.9%, GAN-TSC 69.6%), (Higgs 1k: MUNGE 58.3%, GAN-TSC 59.0%) (MAGIC: MUNGE 0.909, GAN-TSC 0.918) (Evergreen: MUNGE 0.878, GAN-TSC 0.882) ).

## A.3 REAL IMAGES FOR CALTECH-256

The real images for Caltech-256 is shown in Fig. 5.

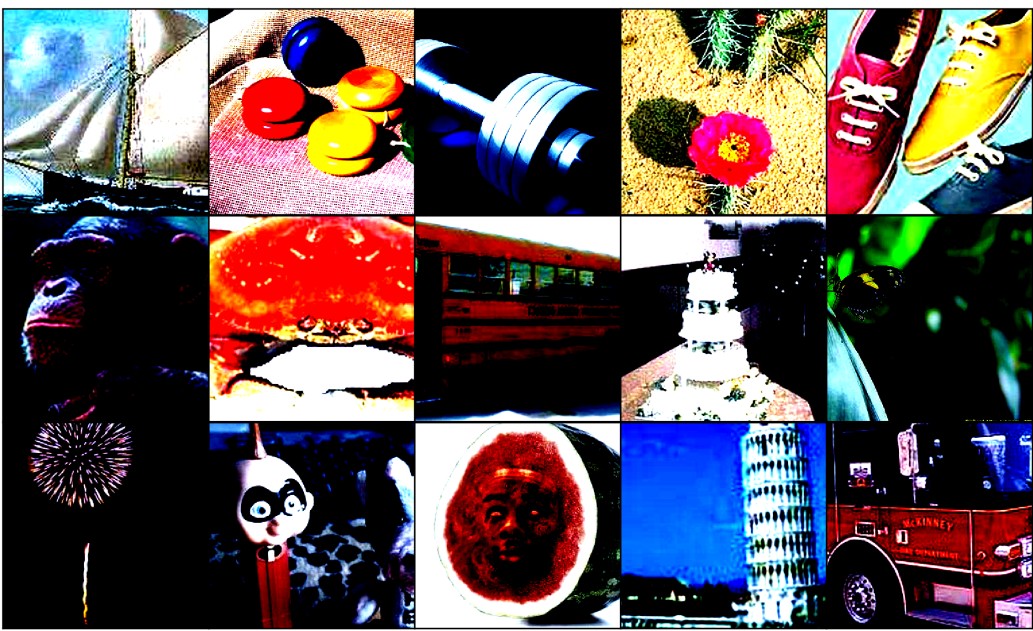

Figure 5: Real images for Caltech-256.

