# OpenReview forum: "Teacher-Student Compression with Generative Adversarial Networks"
_ICLR.cc/2020/Conference — Reject_

### Official Review · AnonReviewer3 · 2019-10-22
**Official Blind Review #3**

**Rating:** 3

**Review:**

This paper proposes an approach for improving teacher-student compression by introducing the assistant of GANs. A conditional GAN is trained for generating synthetic data. Then, the generated data combined with training data is used for knowledge distillation. Experiments on large random forests and deep neural networks demonstrate the effectiveness of the proposed method on data-augmentation. Moreover, an evaluation metric is proposed to evaluate s across-class diversity and intra-class diversity for generative models.

Pros:

+ While using synthetic data of GANs to assist supervised learning has been shown to be failed in pervious works (e.g. [1] and also shown in this paper, this paper presents a new perspective to utilize GAN as a successful data-augmentation technique in teacher student paradigm.
+ Experiments are conducted in several settings including different models (random forests, deep neural networks) and different datasets (images and tabular) to show the effectiveness of proposed method in various settings.
+ The proposed GAN-TSC can be combined with standard augmentation to achieve higher performance as shown in the experiments.
+ This paper is well written and easy to follow.

Cons:

- Knowledge distillation, as a classical model compression technique, has been applied in deep convolutional models for several years. The CIFAR-10 dataset is too simple to evaluate this kind of methods. The author should try to conduct experiments on large scale datasets such as ImageNet, unless the reliability of the proposed algorithm would be very limited.
- The proposed TSCScore seems to be similar with [1], especially when its novelty mainly lies in intra-class diversity compared with IS. It’s necessary to discuss difference between TSCScore and [1].

[1] Konstantin Shmelkov, Cordelia Schmid, and Karteek Alahari. How good is my GAN? ECCV 2018.


**Experience Assessment:**

I have published in this field for several years.

**Review Assessment: Checking Correctness Of Derivations And Theory:**

N/A

**Review Assessment: Checking Correctness Of Experiments:**

I carefully checked the experiments.

**Review Assessment: Thoroughness In Paper Reading:**

I read the paper thoroughly.

---

### Official Review · AnonReviewer2 · 2019-10-23
**Official Blind Review #2**

**Rating:** 3

**Review:**

In this manuscript, authors adopt GAN for data augmentation to improve the performance of knowledge distillation. My concerns are as follows.
1.	The novelty is limited. Using GAN for data augmentation is not new and authors only introduce it for KD, which didn’t address the essential problem of KD itself.
2.	The experiments are not sufficient. For DNNs, authors only compare the performance on CIFAR-10 with the conventional KD. More data sets and benchmark algorithms are helpful to illustrate the effectiveness of GAN data.


**Experience Assessment:**

I have read many papers in this area.

**Review Assessment: Checking Correctness Of Derivations And Theory:**

I assessed the sensibility of the derivations and theory.

**Review Assessment: Checking Correctness Of Experiments:**

I assessed the sensibility of the experiments.

**Review Assessment: Thoroughness In Paper Reading:**

I read the paper at least twice and used my best judgement in assessing the paper.

---

### Official Review · AnonReviewer1 · 2019-10-24
**Official Blind Review #1**

**Rating:** 3

**Review:**

This paper presents an algorithm to generate images for training a student network through distillation. The paper claims the use of the same training data is not necessarily beneficial when it comes to improving the accuracy of the student being trained.
The paper sits on the empirically driven side with sufficient experiments in the context of random forest and CNN.
As a second contribution, the paper proposes a scoring process to evaluate the quality of datasets generated by GAN methods. There are little experiments on this side.


Pros:
- I like the paper and the idea behind being able to improve or even train a student network when the original data is not present.
- Metrics tailored to the problem are relevant.

Negs:

- While there are plenty of experiments, there is a lack of detailed descriptions.
- The scoring for GANS seems to be barely tested. In the scoring GAN, The fact that TSCS drops is enough to be a valid metric (or better than existing?). In the TSC vs inception, it is hard for me to see the unrealistic artifacts and, according to the text, that is not what TSCS is measuring, right? What is the influence of using different student-teacher configuration? (on the time to produce the scores the paper claims NiN and LeNet, is there any difference if using other architectures (ResNet family for instance)? At least, in that case, the time changes. It would be nice to see the stability of this metric to demonstrate that the need for training an inexpensive model is correct.


For the TSC vs Inception, the GANs are subjectively assessed, isn't it (as to select well-trained to Inferior). Would it be possible to see exactly the same images between the two of them? Seems like the difference in quality for IS is significantly larger than for the proposed metric (even in the last gan there is a slight increase in the metric).

It seems to me that IS is just a quality metric based on how a single image looks like. Would it be possible to disentangle the training and the scoring in the proposed metric? What if my hyperparameters for doing the single epoch are totally wrong?


In the general idea of the GAN, while I like it, there is little about how the GAN is actually trained. I guess this GAN is trained using some sort of real data and therefore, the comparison is not totally fair. How many images were used to train this GAN? What would happen if those images are used directly in the distillation framework?

How many images are generated in section 4? Is the influence of pfake related to the dataset? If I have to train using another dataset, how do i set that parameter?

**Experience Assessment:**

I have read many papers in this area.

**Review Assessment: Checking Correctness Of Derivations And Theory:**

I assessed the sensibility of the derivations and theory.

**Review Assessment: Checking Correctness Of Experiments:**

I assessed the sensibility of the experiments.

**Review Assessment: Thoroughness In Paper Reading:**

I read the paper at least twice and used my best judgement in assessing the paper.

---

### Decision · Program_Chairs · 2019-12-19

**Decision:**

Reject

**Comment:**

This paper uses GAN for data augmentation to improve the performance of knowledge distillation.

Reviewers and AC commonly think the paper suffers from limited novelty and insufficient experimental supports/details.

Hence, I recommend rejection.